

# Optical Properties and Aging of Light Absorbing Secondary Organic Aerosol

Jiumeng Liu[1], Peng Lin[2], Alexander Laskin[2], Julia Laskin[3], Shawn M. Kathmann[3], Matthew Wise[4], Ryan Caylor[4], Felisha Imholt[4], Vanessa Selimovic[4‡], John E. Shilling[1,*]

[1] Atmospheric Sciences and Global Change Division, Pacific Northwest National Laboratory Richland, WA, USA.

[2] Environmental Molecular Sciences Laboratory, Pacific Northwest National Laboratory, Richland, WA, USA.

[3] Physical Sciences Division, Pacific Northwest National Laboratory, Richland, WA, USA.

[4] Math and Science Department, Concordia University, Portland, OR, USA.

[‡] Now at Department of Chemistry, University of Montana, Missoula, Montana 59812, USA

*Correspondence to: John E. Shilling (john.shilling@pnnl.gov)

Keywords: Secondary Organic Aerosol, Brown Carbon, Light Absorption





## Abstract

The light-absorbing organic aerosol (OA), commonly referred to as "brown carbon (BrC)", has attracted considerable attention in recent years because of its potential to affect atmospheric radiation balance, especially in the ultraviolet region and thus impact photochemical processes. A growing amount of data has indicated that BrC is prevalent in the atmosphere, which has motivated numerous laboratory and field studies; however, our understanding of the relationship between the chemical composition and optical properties of BrC remains limited. We conducted chamber experiments to investigate the effect of various VOC precursors, $NO_x$ concentrations, photolysis time and relative humidity (RH) on the light absorption of selected secondary organic aerosols (SOA). Light absorption of chamber generated SOA samples, especially aromatic SOA, was found to increase with $NO_x$ concentration, at moderate RH, and for the shortest photolysis aging times. The highest mass absorption coefficients (MAC) value is observed from toluene SOA products formed under high $NO_x$ conditions at moderate RH, in which nitro-aromatics were previously identified as the major light absorbing compounds. BrC light absorption is observed to decrease with photolysis time, correlated with a decline of the organonitrate fraction of SOA. SOA formed from mixtures of aromatics and isoprene absorb less visible and UV light than SOA formed from aromatic precursors alone on a mass basis. However, the mixed-SOA absorption was underestimated when optical properties were predicted using a two-product SOA formation model, as done in many current climate models. Further investigation, including analysis on detailed mechanisms, are required to explain the discrepancy.



## 1. Introduction

Climate forcing by various atmospheric components has been intensely investigated over the last few decades but significant uncertainties still exist (IPCC, 2013). One of the largest uncertainties comes from the role of carbonaceous aerosols, including black carbon (BC) and organic carbon (OC). Black carbon is formally defined as an ideally light-absorbing substance composed of carbon (Petzold et al., 2013) with strong absorption across a wide spectrum of visible wavelengths, which is caused by a significant, wavelength-independent imaginary part k (i.e., ~0.79 (Bond et al., 2013)) of the refractive index. BC has long been known as the strongest light-absorbing aerosol in the visible wavelengths (e.g., Bond et al., 2013). On the other hand, OC has been treated as a scattering species, and only a few recent global modeling studies have focused on the radiative forcing by absorbing OC (Lin et al., 2014a; Feng et al., 2013;Chung et al., 2012). Light absorbing OA are collectively called brown carbon (BrC) (Laskin et al., 2015;Moise et al., 2015;Andreae and Gelencsér, 2006). In contrast to BC, the imaginary refractive index of BrC has stronger wavelength dependence ($\lambda^{-2}$–$\lambda^{-6}$) that increases towards shorter visible and ultraviolet (UV) wavelengths. This broad absorption band in the blue/violet region of the spectrum gives BrC its eponymous yellow or brown color (Alexander et al., 2008;Andreae and Gelencsér, 2006;Lukács et al., 2007). BrC has been widely observed in many environments, including urban environments largely impacted by anthropogenic emissions(Zhang et al., 2013;Du et al., 2014), biomass burning plumes(Lack et al., 2013;Lack et al., 2012;Forrister et al., 2015), over the ocean(Bikkina and Sarin, 2013), rainwater(Kieber et al., 2006) and in the troposphere(Liu et al., 2014;Alexander et al., 2008).



A variety of studies have investigated sources of BrC in both the laboratory and in the field. Incomplete and smoldering combustion of hydrocarbons, especially those associated with biomass burning, is known to directly produce particulate BrC (Hoffer et al., 2006;Hecobian et al., 2010;Lack et al., 2013;Desyaterik et al., 2013;Chakrabarty et al., 2010;Kirchstetter and Thatcher, 2012;Mohr et al., 2013). There is also evidence based on ambient studies of a secondary BrC source (Duarte et al., 2005) and laboratory studies show formation of chromophores (components of molecules that absorb light) through a variety of mechanisms, including photooxidation of aromatics (Lambe et al., 2013;Liu et al., 2015b), ozonolysis of terpenes subsequently aged in the presence of ammonium ions and humidity (Bones et al., 2010;Nguyen et al., 2013;Laskin et al., 2014;Updyke et al., 2012), and a variety of additional aqueous phase reactions, such as lignin (Hoffer et al., 2006) and isoprene oxidation (Limbeck et al., 2003), reactions of carbonyls (e.g., glyoxal, methyglyoxal) in acidic solutions(Sareen et al., 2010), with amino acids (De Haan et al., 2009), amines (De Haan et al., 2009;Powelson et al., 2014;Zarzana et al., 2012), or ammonium salts (Sareen et al., 2010;Lin et al., 2015a;Galloway et al., 2009;Kampf et al., 2012;Shapiro et al., 2009). Among those studies, it is suggested that the chemical and optical properties of laboratory generated SOA might be influenced by a variety of factors, including the composition of the volatile organic carbon (VOC) precursor, oxidation chemistry, relative humidity (RH), and potentially "aging" at longer time scales (i.e., in-particle reactions and photobleaching). Particularly, SOA aged in the presence of dissolved ammonium has been shown to produce BrC efficiently, which may contribute to aerosol optical density in regions with elevated concentrations of ammonium salts (i.e., Updyke et al., 2012).



This study focuses on measuring light absorption by laboratory-generated SOA that simulate both urban
and remote environments. Four VOCs representative of biogenic and anthropogenic emission are
chosen as SOA precursors in this study. Biogenic VOCs selected include isoprene and α-pinene, of
which isoprene is the most abundant biogenic non-methane hydrocarbon emitted into the atmosphere
(Guenther et al., 2006), while α-pinene accounts for approximately 40% of global monoterpene ($C_{10}H_{16}$)
emissions (Guenther et al., 2012). For anthropogenic VOCs, we selected trimethylbenzene (TMB) and
toluene, the photooxidation of which in the presence of $NO_x$ is a major source of anthropogenic SOA
(Ng et al., 2007;Kleindienst et al., 2004;Henze et al., 2008). Four different types of experiments were
conducted to investigate the effects of (1) $NO_x$ levels, (2) VOC precursors, (3) photolysis time, and (4)
RH on SOA light absorption. We compare the light absorption of formed SOA by ultraviolet/visible
(UV/Vis) absorption measured from aerosol samples extracted in water and methanol.

**2. Experimental methods**
Experiments were performed in the indoor 10.6 m$^3$ Teflon chamber at the Pacific Northwest National
Laboratory (PNNL) operating in batch mode where a discrete quantity of a VOC is introduced into the
chamber and allowed to react with the gas-phase oxidants (Liu et al., 2012). The Teflon chamber was
flushed continuously with dry purified air until particle concentrations were less than 5 cm$^{-3}$ prior to all
experiments. For each experiment, a measured amount of VOC was injected into a glass bulb with a
syringe, evaporated with gentle heating, and transferred to the chamber in a flow of purified air. After
the VOC injection, 0.5 mL of $H_2O_2$ solution (Sigma-Aldrich, 50 wt% in $H_2O$) was injected into the
chamber in the same manner. Humidity was controlled by passing pure air at a variable flow rate



through pure water (18.2 MΩ cm, <5ppbv TOC) with a HEPA filter downstream of the bubbler to
remove any contaminant particles. In experiments in which $NO_x$ were present, NO was injected from a
gas cylinder containing a known NO concentration (500 ppm, Matheson Tri-Gas®) with flows regulated
by mass flow controllers. After all components were injected and well-mixed in the chamber, UV lights
were turned on to initiate photooxidation. The UV flux in the chamber, averaged $J_{NO2}$=0.16 min$^{-1}$, was
measured continuously by a radiometer that is calibrated to an equivalent photolysis rate of $NO_2$ and
suspended in the center of the chamber. Measurements of $J_{NO2}$ using the photostationary state method
were in agreement with the radiometer measurements (Leighton, 1961).

During the experiments, a suite of online instruments were used to characterize the gas- and particle-
phase composition. The mixing ratios of the hydrocarbons were continuously monitored with an
Ionicon proton-transfer-reaction mass spectrometry (PTR-MS). The mass loading of the aerosol
particles was measured using an Aerodyne high-resolution time of flight mass spectrometer (HR-ToF-
AMS) (DeCarlo et al., 2006), while the number and volume concentrations were measured with a TSI
scanning mobility particle sizer (SMPS). An NO/NO$_2$/NO$_x$ analyzer (Thermo Environmental
Instruments model 42c) was used to measure the concentration of NO and $NO_x$. A UV absorption $O_3$
analyzer (Thermo Environmental Instruments model 49C) allowed for the measurement of $O_3$
concentration.

SOA samples were collected on filters to measure their light absorption. Photooxidation products were
collected onto PTFE filters (Pall Life Sciences, 47 mm, 1 μm pore size) at a flow rate of 9 L min$^{-1}$ for a



collection period of 60-120 minutes. Typically at least 20 µg of organic mass is required for accurate
measurement of light absorption. As described in previous studies (Hecobian et al., 2010;Zhang et al.,
2011), filters were extracted in high purity water (> 18.2 MΩ cm), filtered through a 25mm diameter
0.45 µm pore syringe filter (Fisher Scientific, Fisherbrand$^{TM}$ Syringe Filters) and transferred into a long-
path (100 cm pathlength) UV-Visible spectrometer (Ocean Optics) to determine the light-absorption
spectra. After water extraction, filters were also sonicated in methanol (VWR International, A.C.S.
Grade) to extract non-water soluble mass (Liu et al., 2013;Liu et al., 2015a). Total absorption due to
BrC (Abs(λ)) is determined as the sum of water-soluble and methanol-extracted absorption from the
sequential extraction processes. An extraction efficiency test was performed with 6 filters, in which
filters were cut in halves, one half extracted with methanol only and the other half processed with the
sequential extraction. Results show that the sum of light absorption from the sequential extraction is
comparable to methanol extraction alone, with a slope within 8% of 1 (Figure S1).  Studies have shown
that the extraction efficiency of organic mass is >90% using methanol as the solvent (Chen and Bond,
2010;Updyke et al., 2012). Thus, it is reasonable to assume that total light absorption determined from
the sequential extraction procedure closely approximates the "true" optical properties of the SOA
samples. The limit of detection (LOD) was 0.081 Mm$^{-1}$ in the 300-700 nm wavelength range with an
estimated uncertainty of 21%. The mass absorption coefficient (MAC) was then estimated using
equation 1:
$$MAC(\lambda) = \frac{Abs(\lambda)}{OM} \tag{1}$$

in which Abs(λ) is the light absorption from filter-collected aerosol samples at a wavelength λ, and OM
is the SOA mass concentrations on the filter estimated from AMS measurements and the sampled air



volume. Wall-loss corrections were not applied to either measured SOA mass concentrations or light
absorption determined from filter-collected aerosol samples for consistency. Based on lowest SOA mass
concentrations during all experiments, the LOD of the MAC is estimated as 0.004 $m^2\ g^{-1}$.

**2.1 Description of the SOA two-product model**
Ambient studies have shown that SOA produced from urban emissions in isoprene-rich environments
tend to have much lower BrC absorption compared to that in anthropogenic emission dominant
environments (Zhang et al., 2011). In our study, two mixed precursor experiments were conducted to
investigate the changes in aromatic BrC due to addition of isoprene reaction products. We employ a
two-product model to describe the partitioning of organic mass between aromatic- and isoprene-derived
SOA (Pankow, 1994;Odum et al., 1996). SOA yield parameters for pure compounds are determined by
fitting real-time batch mode data as described in the literature (Presto and Donahue, 2006). In the mixed
precursor experiments, the PTR-MS data is used to determine the amount of each precursor reacted
during the filter collection periods. Then, the pure compound yield parameterizations are used to
calculate the relative fractions of the isoprene- and aromatic-derived SOA collected on the filter. The
calculation assumes that all SOA components are mutually miscible and reproduced the measured SOA
mass with the difference less than 10% (Table S1). These fractions are then used along with the optical
properties of the single-precursor SOA to predict the optical properties of the mixed aerosol.

**3. Results and Discussion**
3.1 Effects of VOC types and $NO_x$ levels



The wavelength-dependent MAC values for SOA derived from four selected precursor VOCs are
plotted in Figure 1. In general, the shapes of the spectra are characteristic of typical atmospheric BrC
materials, with relatively higher absorption in the UV range (i.e., Hecobian et al., 2010;Chen and Bond,
2010). Figure 2 shows a comparison of the MAC at 365 nm among four different SOA samples
(isoprene, α-pinene, TMB and toluene) produced under $NO_x$-free and high-$NO_x$ conditions.

The MAC values of isoprene SOA are close to the LOD in the 300-700 nm wavelength range and there
is no significant difference in the UV-Vis spectra of isoprene SOA formed under $NO_x$-free and high-
$NO_x$ conditions. Quantum mechanical calculations suggest that electrons must be delocalized over the
equivalent of 7-8 bond lengths before an absorption will occur at 360 nm (Kuhn, 1949). Therefore our
results suggest SOA produced from isoprene photochemical oxidation does not contain products that
have extended carbon conjugated chains, consistent with current understanding that isoprene
photochemical oxidation products consist of carbonyls, hydroxycarbonyls, diols and organic peroxides
(e.g., Nguyen et al., 2011). On the other hand, Lin et al. (2014) has suggested that acidic seeds may
promote formation of oligomers through reactive uptake of IEPOX and produced light-absorbing
organic aerosols under certain conditions (Lin et al., 2014b). In our experiments, neither acidic seeds
nor excess ammonia are present, which likely explains the difference between our observations and
those of Lin et al. (2014).

SOA formed from photochemical oxidation of α-pinene also showed very limited light absorption.
However, we observe a slight increase in the MAC values at wavelengths below 450 nm for the α-





pinene SOA formed under high-NO$_x$ conditions relative to that formed in the absence of NO$_x$. These
observations are consistent with other studies that have found minimal light absorption for α-pinene
SOA, again indicating that the compounds partitioning to the condensed phase do not have extended
conjugation (Henry and Donahue, 2012;Nakayama et al., 2010;Laskin et al., 2014).

In contrast to the SOA produced from the terpene precursors, aromatic precursors representative of
anthropogenic VOCs produce SOA that significantly absorbs light, particularly in the UV wavelength
range. Overall, the MAC values of the SOA produced from both TMB and toluene are much higher than
biogenic SOA, for both NO$_x$-free and high-NO$_x$ conditions (Figure 2). Lambe et al. (2013) has
suggested that the conjugated double bonds retained in oxidation products of aromatic precursors are
likely to contribute to absorption in the ultraviolet to near visible range (Lambe et al., 2013). SOA
formed from non-aromatic precursors, on the other hand, did not show strong light absorption in the
ultraviolet/visible range due to lack of extended conjugated double bond networks.

For both toluene and TMB SOA, high NO$_x$ products show substantially higher light absorption than low
NO$_x$. Shown in figures 1 and 2, aromatic SOA formed under high NO$_x$ conditions have much higher
MAC values, both in the UV and in the visible. Several studies, based upon both chamber and field
observations, have suggested that nitrogen-containing molecules are strong light absorbers (i.e.,
Nakayama et al., 2013;Liu et al., 2015b;Zhang et al., 2011;Lin et al., 2015b). In a companion study, we
reported detailed characterization of the most prominent BrC chromophores in toluene-SOA formed
under both NO$_x$-free and high-NO$_x$ conditions by deploying liquid chromatography combined with a



UV/vis detector and high-resolution mass spectrometry (LC-UV/Vis-ESI/HRMS) (Lin et al., 2015b).
Samples of toluene-SOA produced under high-$NO_x$ and $NO_x$-free conditions have substantially different
chemical compositions. In high-$NO_x$ SOA, we identified 15 nitro-aromatic compounds, including
nitrocatechol, dinitrocatechol and nitrophenol, the total absorbance of which accounts for 60% and 41%
of the overall absorbance in the wavelength ranges of 300-400nm and 400-500nm, respectively (Lin et
al., 2015b). In contrast, photooxidation products observed in $NO_x$-free SOA are dominated by non-
aromatic compounds with high degree of saturation, which did not show substantial light absorption in
the UV/Vis range. Similar to toluene SOA, TMB SOA produced under high-$NO_x$ conditions contains
nitrogen-containing compounds in contrast to $NO_x$-free SOA, which explains the difference in light-
absorbing properties (Liu et al., 2012).

For similar reaction conditions, the TMB-derived SOA are less absorptive than the toluene SOA. The
difference in the light absorption properties between toluene SOA and TMB SOA may be explained by
the difference in the production of nitrophenols. Sato et al. (2012) showed that nitrophenols were not
detected in the TMB SOA, possibly due to the fact that $NO_2$ addition to the phenoxy radical formed in
reaction of TMB with OH is inhibited (Sato et al., 2012). Our measurement is consistent with this
hypothesis and infers that nitro-aromatics such as nitrophenols are the main sources of light absorption
for the aromatic SOA.

The MAC values of SOA produced from aromatic VOCs are comparable to those of other light-
absorbing material relevant to atmospheric aerosol particles, such as fulvic acid. Shown in Figure 3a,





the blue shaded area represents the measured MAC range of SOA produced in the toluene+NO$_x$
experiments, with the MAC of Suwannee River fulvic acid as a reference. Over the wavelength range
380-480 nm, toluene SOA has higher MAC values than fulvic acid. Since fulvic acid is often cited as a
surrogate of atmospheric BrC, this comparison shows that light absorption by BrC produced from
anthropogenic VOCs can be significant under certain photochemical condition.

3.2 Mixed precursor experiments
Results from laboratory studies have shown that the addition of isoprene reduced the BrC absorption of
aerosols formed from toluene+α-pinene mixtures(Jaoui et al., 2008). The lower absorption was
attributed to decreased organic aerosol yields (e.g., lower amounts of light-absorbing SOA were formed)
(Jaoui et al., 2008). From ambient observations, Zhang et al. (2011) reported contrasting light
absorption properties in two urban environments. Fresh SOA in LA displayed much higher light-
absorption presumably because of the anthropogenic-dominated environment, while Atlanta aerosols
formed from a mix of anthropogenic and biogenic (isoprene) VOC precursors had a 4-6 times lower
MAC value (Zhang et al., 2011). Hecobian et al. (2010) measured the light absorption of water-soluble
organic carbon (WSOC) in Atlanta in different seasons and found that the winter WSOC has a ~3 times
higher MAC than summer, due to biomass burning influence in winter (Hecobian et al., 2010). Using
summer-time samples collected in Atlanta, Liu et al. (2013) reported a significantly higher BrC MAC
value that was associated with primary anthropogenic emissions, compared to the lower MAC value
observed at sites with local anthropogenic emissions on top of regional biogenic-dominant emissions
(Liu et al., 2013). To investigate whether isoprene photooxidation products enhance or inhibit



absorption of aromatic SOA, we conducted two mixed-precursor experiments. Figure 4 shows the
comparison of MAC values at 365 nm of SOA formed from single precursor and from mixed isoprene
and aromatic VOCs, under high-$NO_x$ conditions. In both isoprene/toluene and isoprene/TMB
experiments, the SOA formed has lower MAC values than those formed from the pure aromatics alone.
Qualitatively, this is the behavior that one would expect, since non-absorbing isoprene SOA will "dilute"
the chromophores from the aromatic-derived SOA. To determine whether the total aerosol absorption
can be described quantitatively, we first estimate the mass of aromatic- and isoprene-derived SOA
(Table S1) using a partitioning model described in section 2.1. We then calculate predicted aerosol
MAC values as the mass-weighted average of the MAC values measured for the pure isoprene- and
aromatic- derived SOA species. Figure 4 shows a comparison of the measured and predicted mixed-
precursor SOA optical properties. The predicted MAC values are 31%-55% lower than the
measurements, a difference that is likely outside of the measurement uncertainty. There are several
potential explanations for the difference between the predicted and observed MAC values. First, it is
possible that SOA formation is not well-described by partitioning theory. One potential source of error
in our calculation is that we assume isoprene and aromatic SOA are fully miscible in one another;
however, we note that the total predicted SOA mass is within 10% of the observed SOA mass and hence
the underprediction of the MAC values cannot be explained by this error. A second possibility is that
the partitioning model underestimates the mass of aromatic SOA that has condensed into the mixed-
phase particles. Studies have shown that gas-phase wall loss of toluene reaction products can be
significant under certain conditions in batch-mode experiments (Zhang et al., 2014). The SOA yield
parameterizations are based on data collected in the absence of seed particles, in which case gas-phase



wall loss could be significant. However, isoprene reacts much more quickly than toluene (Figure S2);
therefore isoprene SOA should form first and provide surface area which should mitigate gas-phase
wall loss of the toluene reaction products. Because no seed particles were present in the pure toluene
experiments, we would expect those yield values to be biased low relative to the toluene yield in the
mixed precursor experiments, thus potentially explain the underprediction of MAC values. A third
possibility is that reactions between organic peroxide and alcohol functionalities known to be the
dominant component of isoprene SOA (Krechmer et al., 2015) react with toluene SOA components to
produce oligomers capable of absorbing in the UV/VIS. Examination of the AMS spectra in the mixed
experiments and comparison to the spectra of the pure aromatic- and isoprene- SOA were inconclusive
in providing evidence of this hypothesis. Samples were not collected for detailed analysis by LC-
UV/Vis-ESI/HRMS. Therefore, at this time we can't conclusively explain the apparent absorption
enhancements we observe.

3.3 Effect of Relative Humidity on Light Absorption by aromatic SOA
In order to investigate the effect of RH on SOA light absorption, both toluene and TMB photo-oxidation
experiments were conducted at fixed VOC and $NO_x$ values but variable RH levels (Table 1). Figure 5
illustrates the light absorption spectra of toluene- and TMB-derived SOA as a function of experimental
RH. The data shown here were from samples collected at a photolysis time of 200 minutes which
corresponds to the time when light absorption reached its highest value. In both TMB and toluene
experiments, SOA generated under dry conditions (RH <5%) displayed significantly lower MACs than
SOA formed at RH>30%. SOA formed at 30, 50 and 80% RH have similar light absorption to one



another. Thus moderate RH enhances the MAC values by a factor of 1.33 at 365 nm and further
increases in RH have no effect. An overview of toluene-SOA molecular compositions was analyzed by
nano-DESI/HRMS(Lin et al., 2015b), and showed that a large number of nitrogen containing
compounds (CHON) were produced under moderate RH condition (Figure S3). The difference in
molecular compositions suggest that low RH inhibited the formation of nitrogen-containing compounds,
which have been shown to be major light absorbers in toluene-SOA formed in the presence of
$NO_x$(Nakayama et al., 2013;Liu et al., 2015b;Zhang et al., 2011;Lin et al., 2015b).

We are unable to identify any gas-phase reactions in the toluene photolysis mechanism directly
involving water vapor. Thus, we conclude that RH must be affecting particle-phase reactions that
enhance chromophore formation. Several studies have investigated the effect of RH on various particle-
phase SOA chemistry and optical properties. Song et al. (2013) found that SOA produced from α-
pinene+$NO_x$+$O_3$ in the presence of acidic seed aerosols at elevated RH was less light-absorbing than
SOA formed under dry conditions (Song et al., 2013) , which is opposite of our observations. They
suggested that the change in light-absorbing properties might be triggered by evaporation of water,
which may have enhanced the acidity of aerosol seeds (Nguyen et al., 2012), thererby promoting
oligomerization reactions. Zhong et al. (2014) investigated the light absorption of BrC formed from
wood burning and observed a faster decay of chromophores at higher RH, which they attributed to the
decomposition of chromophores by $H_2O_2$ that is produced by aqueous-phase photooxidation in the
presence of elevated water content level (Zhong and Jang, 2014). Moderate to high RH may promote
heterogeneous reactions, which aids in the reactive uptake of volatile compounds into aerosols. Cao and



Jang (2010) decoupled SOA mass into partitioning and heterogeneous aerosol production in a toluene-
NO$_x$ system, and suggested that moderate RH results in a higher fraction of SOA formed via
heterogeneous reactions than low RH conditions(Cao and Jang, 2010). Similar effects might be also
pertinent to the toluene SOA. Another possible explanation is that SOA formed under low RH
conditions may exist in a viscous, semi-solid, or glassy state due to particle-phase oligomerization
reactions (Saukko et al., 2012;Shiraiwa et al., 2013) while SOA formed at moderate/high RH would be
less viscous. Since only one experiment was conducted under dry condition for each compound it is
difficult to draw conclusions, but further investigations are warranted.

3.4 Effect of photochemical aging on light absorption of aromatic SOA
Atmospheric aerosols have a wide range of lifetimes, ranging from hours to days (i.e., Wagstrom and
Pandis, 2009). Previous studies have observed a decrease in aerosol absorption with aging in BrC from
various sources including biomass burning and SOA formed from aromatics (Forrister et al.,
2015;Zhong and Jang, 2011;Lee et al., 2014).  We therefore performed several experiments to study the
effect of aging on BrC absorption. Figure 6 shows the MAC values at 365 nm as a function of
photolysis time for toluene and TMB SOA produced in the presence of NO$_x$ at 30% RH. We observe a
clear decrease in aerosol absorption with aging with MAC values decreasing by ~35% after 400 minutes
and >50% after 800 minutes.

Laboratory studies have suggested that photo-bleaching was due to degradation of BrC chromophores
(Lee et al., 2014;Zhong and Jang, 2011;Zhong and Jang, 2014). In our observations, the decrease of



MAC with aging is accompanied by a decreasing ON-to-OM ratio, shown in Figure 6. Here we define
the term ON as the sum of NO, $NO_2$ and $C_xH_yO_zN_w$ families measured by AMS, to represent organic
nitrates formed during the experiments. NO and $NO_2$ come exclusively from organic nitrates in these
experiments. Ammonium is below the instrument detection limit, and the ratio of m/z 30:46 (around 5-6)
is indicative of organic nitrate, thus ruling out formation of ammonium nitrate (Farmer et al., 2010).
Therefore, the decrease in the aerosol ON:OM with time indicates the loss of ON groups (Figure 6). ON
groups have been identified as the strong light absorbers in aromatic SOA formed under high-$NO_x$
conditions, thus the relative decrease in ON fraction relative to OM is consistent with the observed
evolution in OA light absorption.

This observed loss of ON could be caused by photolysis and/or hydrolysis of ON groups. Lee et al.
(2014) has observed a substantial decline in the double bond equivalent (DBE) values upon photolysis
of aromatic SOA, and suggested that the decrease in SOA light absorption and chemical composition
was due to photolysis (Lee et al., 2014). On the other hand, Liu et al. (2012) suggested that particle-
phase hydrolysis could substantially reduce ON group concentration, which they also related to a
decrease in BrC light absorption (Liu et al., 2012). To distinguish between the effects from photolysis
and hydrolysis, SOA was allowed to age in the chamber with UV lights off but at elevated RH in
several experiments. Shown in figure 7, the MAC values of toluene and TMB SOA are approximately
constant with aging despite the elevated RH. Therefore, we conclude that decrease in MAC values are
driven primarily by photolysis (i.e., photobleaching), which is correlated with loss of ON groups that
have been shown in many studies, including our sister study, to be BrC chromophores (Lin et al.,



2015b;Liu et al., 2015b;Zhang et al., 2013). The effect of RH is less clear, with the dark experiments
suggesting the net effect of water-related processes, such as hydrolysis and oligomerization, is either
negligible or tends to slightly enhance BrC light absorption, while comparison of experiments with and
without RH (section 3.3) suggesting moderate RH enhances the SOA MAC values.

3.5 Imaginary refractive indices
So far, our discussion focused on mass-normalized absorption based on solution measurements, which
is not directly relatable to light absorption by aerosol particles. Therefore, we derive the imaginary
refractive index, $k$, from spectroscopic data, which can be incorporated into climate models. The $k$ value
is derived using equation 2:
$$k = \frac{\rho_p \lambda \cdot Abs(\lambda)}{4\pi \cdot OM} = \frac{\rho_p \lambda}{4\pi} MAC(\lambda) \qquad (2)$$

where $Abs(\lambda)$ is the solution absorption at a given wavelength, OM is the organic mass extracted in
solution, and $\rho_p$ is the density of organic aerosols. The density of organic aerosols was calculated by
comparing the volume-weighted mobility size measured by SMPS and the mass-weighted aerodynamic
size distribution measured by AMS (DeCarlo et al., 2004). A density of $1.25\pm0.3$ g cm$^{-3}$ was obtained
for SOA produced under NO$_x$-free conditions, while a density of $1.41\pm0.2$ g cm$^{-3}$ was estimated for
SOA produced in high-NO$_x$ experiments. Those density values were employed in equation 2 to estimate
$k$ values at 365 nm for various types of SOA, which are summarized in Table 2 (k values for the 300-
700 nm range are listed in Table S2).



Although α-pinene and isoprene have large contribution to the global SOA budget, they were shown to
produce SOA with very small light absorption coefficients, which agrees with literature data (i.e.,
Nakayama et al., 2010;Lang-Yona et al., 2010). The SOA compounds produced are dominated by
carbonyl, carboxyl, and hydroxyl functional groups, which do not have strong electronic transitions in
the UV/Vis range. As a result, those biogenic SOA particles are expected to have a mostly cooling
effect on global radiative balance. However, studies have shown that biogenic SOA can be converted
into BrC via reactions with dissolved ammonia (Updyke et al., 2012;Laskin et al., 2014). Furthermore,
it has been demonstrated that reactive uptake of IEPOX into acidic aerosols produce BrC (Lin et al.,
2014b), which may have substantial impacts on specific regions with elevated ammonia levels and/or
active IEPOX chemistry.

In the present study, the SOA generated from the photooxidation of aromatic VOC precursors,
particularly toluene, were found to have significant absorption in the UV/Vis range when formed in the
presence of $NO_x$. Toluene-SOA formed under high-$NO_x$ conditions has a $k$ value ranged from 0.019 to
0.047 at 365 nm, and 0.011-0.033 at 405 nm. Shown in Figure 3b, the k values are in good agreement
with the measurement by Nakayama et al. (2010), where reported k values were 0.047 at 355 nm and
0.007 at 532 nm(Nakayama et al., 2010). The k values reported by Zhong and Jang (2011) and Liu et al.
(2015) are close to the lower limit from this work, the former reported a k value of 0.0214 at 350 nm,
and the latter reported a range of 0.022-0.033 at 320 nm(Zhong and Jang, 2014;Liu et al., 2015b).
However, the $k$ values derived in this work are substantially higher than those in Nakayama et al. (2013),
which reported $k$ values ranging from 0.0018 to 0.0072 at 405 nm. A possible explanation is the



difference in NO$_x$ levels among the experiments; Zhong and Jang (2011) and Nakayama et al. (2013)
studies were conducted at NO$_x$ levels lower than 1 ppmv(Zhong and Jang, 2014;Nakayama et al., 2013),
which are lower than employed in our study. Nakayama et al. (2013) has reported that light absorption
of SOA has a dependence on NO$_x$, that MAC increases with NO$_x$ (Nakayama et al., 2013), which likely
also explains the higher k values reported by earlier work from the same group (Nakayama et al., 2010).
Another potentially important difference among the experiments is the RH, with Nakayama 2013 and
the Liu studies conducted under dry conditions(Nakayama et al., 2013;Liu et al., 2015b). From what we
have observed, moderate RH could enhance the light absorption of BrC.

**4. Conclusions and Atmospheric Implications**
Among ambient studies reporting BrC light absorption, high MAC values are almost exclusively
reported for aerosols attributed to biomass burning (Kirchstetter et al., 2004;Hoffer et al.,
2006;Alexander et al., 2008;Dinar et al., 2008;Chakrabarty et al., 2010;Lack et al., 2013), and the
limited number of models that include BrC generally use biomass burning aerosol optical properties as
high-absorption references (Lin et al., 2014a;Feng et al., 2013). Our results suggest that organic aerosols
formed from certain anthropogenic VOC precursors also display efficient light absorption. Specifically,
the MAC values obtained from the toluene+high-NO$_x$ experiment were comparable to that of fulvic acid,
which has been used as model compounds for biomass burning HULIS(Dinar et al., 2006;Brooks et al.,
2004;Chan and Chan, 2003;Fuzzi et al., 2001;Samburova et al., 2005). The results suggest that in
addition to BrC from biomass burning, the photooxidation of anthropogenic precursors can also have
significant impacts on light absorption at wavelengths that drive photochemical reactions.




BrC observed in urban environments has large variations in reported MAC values, and our mixed-
precursor experiments may provide some explanations for the observed variation. From our
measurements, SOA formed from mixtures of isoprene+aromatic VOC have lower MAC values than
those formed from the pure aromatics, suggesting that isoprene photooxidation products dilute light-
absorbing compounds. Therefore, it is possible that some of the variance in BrC properties between
urban sites can be explained by the presence or absence of biogenic emissions. In addition, our results
suggested that $NO_x$ concentration, RH level, and photolysis time have considerable influences on the
formation and decay of light-absorbing compounds. The result suggests that we should revisit how SOA
is treated in climate models, especially in urban areas. Several current regional and global models
include $NO_x$-dependent SOA yield (Lane et al., 2008;Farina et al., 2010;Ahmadov et al., 2012);
accurately parameterizing BrC formation from SOA will require a similar strategy.

**Acknowledgements**
Authors acknowledge support by the Laboratory Directed Research and Development funds of Pacific
Northwest National Laboratory (PNNL). A portion of this study was performed at the William R. Wiley
Environmental Molecular Sciences Laboratory, a national scientific user facility sponsored by the
DOE's Office of Biological and Environmental Research and located at PNNL. PNNL is operated for
the U.S. Department of Energy by Battelle Memorial Institute under Contract No. DE-AC06-76RLO

433 1830.




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



Table 1. Summary of experiments and experimental conditions described in this work.

| Experiment | Experiment type | VOC | Initial VOC concentration (ppb) | Initial NO (ppb) | RH (%) |
|---|---|---|---|---|---|
| 1 | 1 | isoprene | 359.37 | <1 | 30 |
| 2 | 1 | α-pinene | 22.73 | <1 | 30 |
| 3 | 1 | TMB | 316.30 | <1 | 30 |
| 4 | 1 | toluene | 339.92 | <1 | 30 |
| 5 | 2 | isoprene | 311.45 | 1754.67 | 30 |
| 6 | 2 | α-pinene | 45.45 | 466.09 | 30 |
| 7 | 2 | TMB | 289.94 | 1589.6 | 30 |
| 8 | 2 | toluene | 317.26 | 1800 | 30 |
| 9 | 2 | Isoprene+TMB | 178.51+123.71 | 1800 | 30 |
| 10 | 2 | Isoprene+toluene | 158.09+106.43 | 1800 | 30 |
| 11 | 3 | TMB | 263.58 | 1500 | 30 |
| 12 | 3 | toluene | 339.92 | 1900 | 30 |
| 13 | 4 | TMB | 263.58 | 1800 | <5 |
| 14 | 4 | TMB | 263.58 | 1800 | 50 |
| 15 | 4 | TMB | 263.58 | 1800 | 80 |
| 16 | 4 | Toluene | 396 | 1800 | <5 |
| 17 | 4 | Toluene | 300 | 1800 | 50 |
| 18 | 4 | Toluene | 339.92 | 1800 | 80 |





Table 2. Derived imaginary part of refractive index (k) of brown carbon formed from various VOC
precursors at 365 nm. Tabulated values are $k \times 10^3$.

|  | NO$_x$-free | High-NO$_x$ |
| --- | --- | --- |
| Isoprene | 0.029 | 0.196 |
| α-pinene | 0 | 1.15 |
| TMB | 0.967 | 6.028-9.899 |
| toluene | 0.461 | 19.48-46.87 |





Figure 1. MAC values for SOA formed under $NO_x$-free and high-$NO_x$ conditions, from isoprene, α-pinene, TMB, and toluene. Note the 10× difference in scale between the terpene and aromatic precursors.






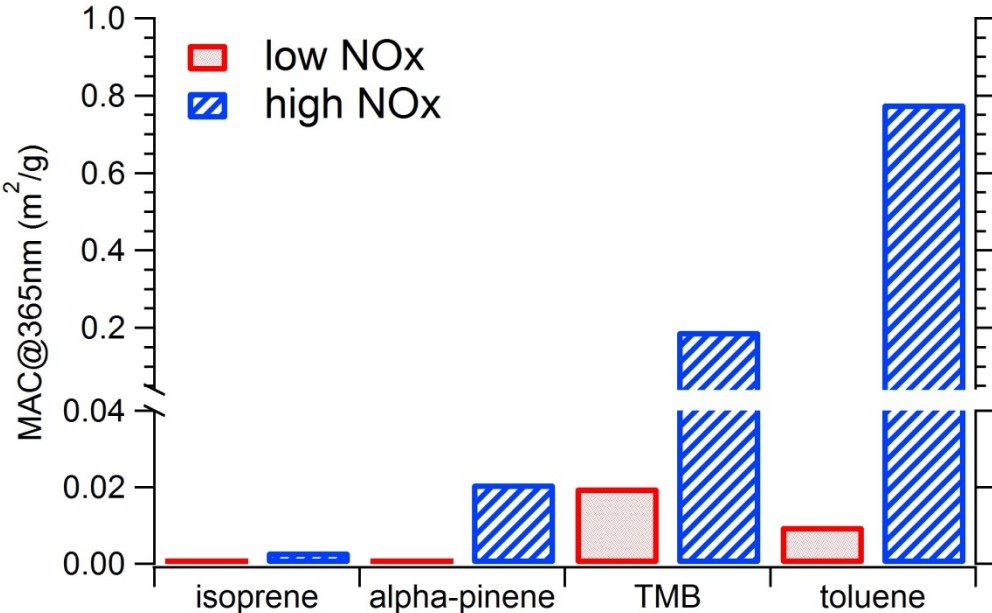


Figure 2. Comparison of MAC from various types of SOA, at a wavelength of 365 nm.


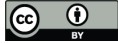

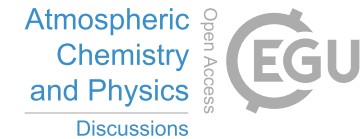

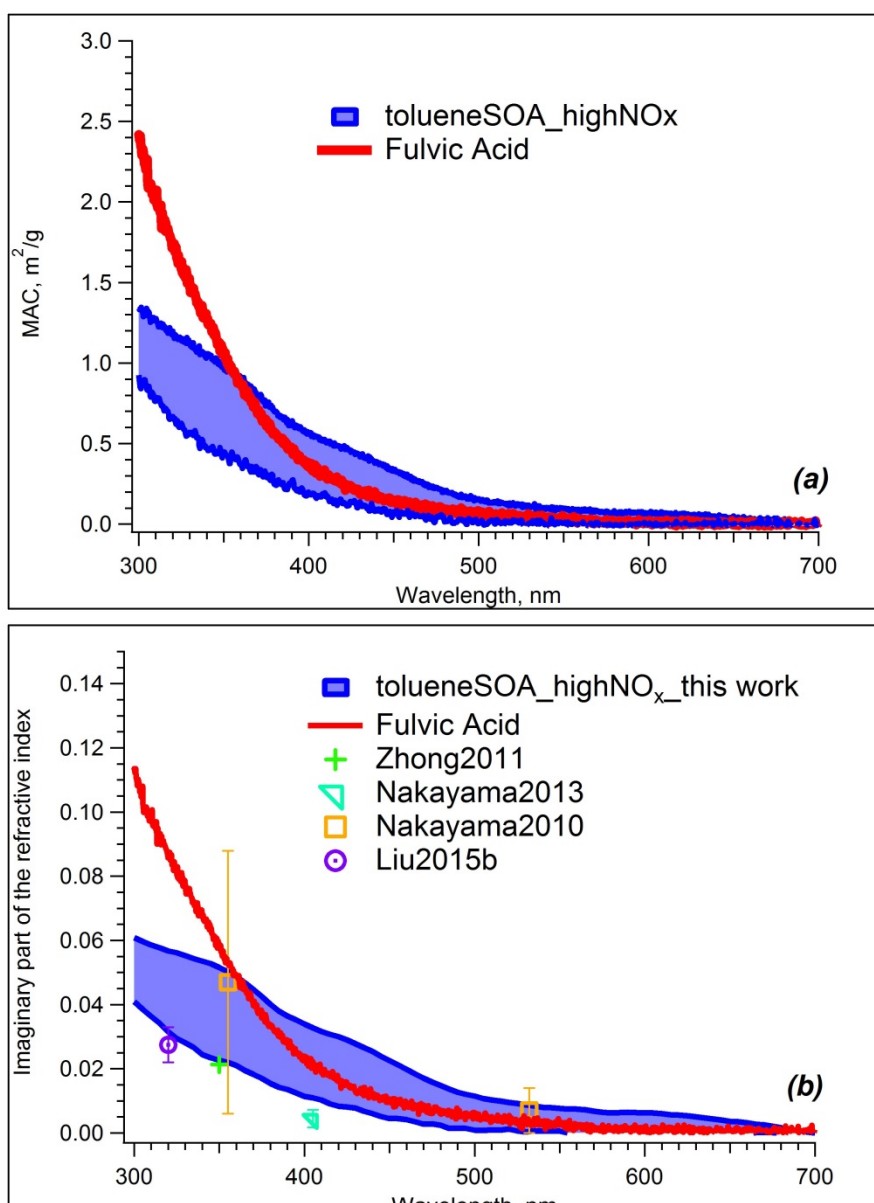


Figure 3. (a) MAC values of Suwanee River fulvic acid (SRFA), and toluene-SOA formed at different
high-NO$_x$ conditions. (b) Imaginary part of the refractive index, k, derived from toluene high-NO$_x$ SOA
measurements through the 300-700 nm range, with SRFA and literature data as references (Nakayama
et al., 2010;Nakayama et al., 2013;Liu et al., 2015b;Zhong and Jang, 2011). SRFA k values were
estimated assuming a density of 1.47 g cm$^{-3}$ (Dinar et al., 2006).

718





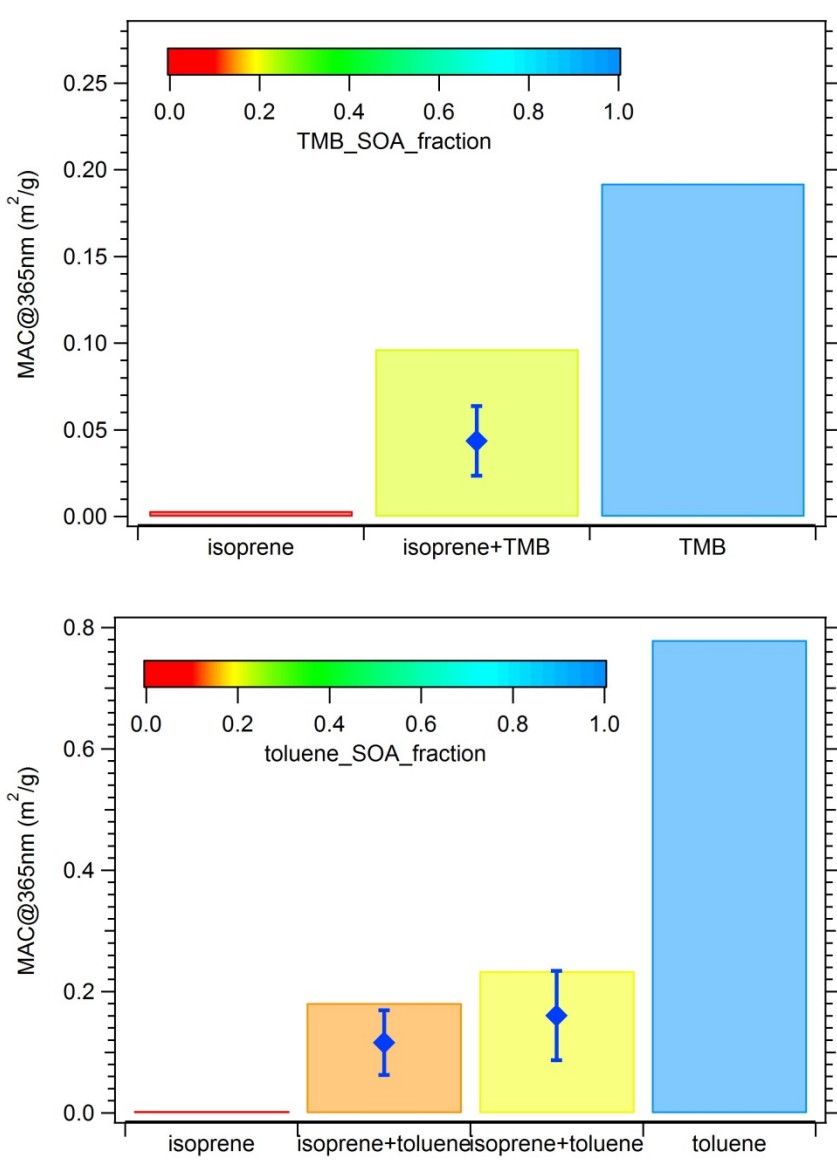

719

Figure 4. Comparison of MAC values from single-precursor and mixed precursor experiments. Bars
represent the MAC values at 365 nm from measurements, and are color-coded by the mass fraction of
aromatic SOA. The blue diamonds represent the predicted MAC values based on the modeled fraction
of isoprene SOA and aromatic SOA, with error bars indicating the uncertainty.





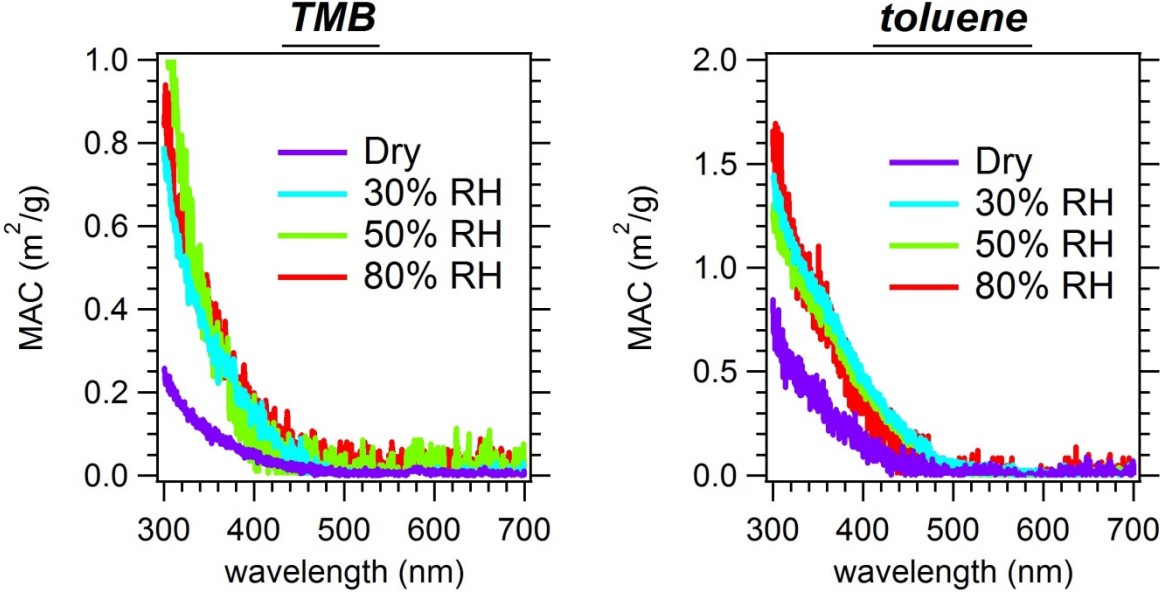


Figure 5. MAC spectra of TMB and toluene SOA formed at <5%, 30%, 50% and 80% RH.






Figure 6.Measurements of the MAC values (at 365 nm) of toluene and TMB SOA formed at 30% RH in

the presence of $NO_x$ as a function of photochemical age (top panels). The bottom panels show the
AMS-measured ON-to-OM ratio.





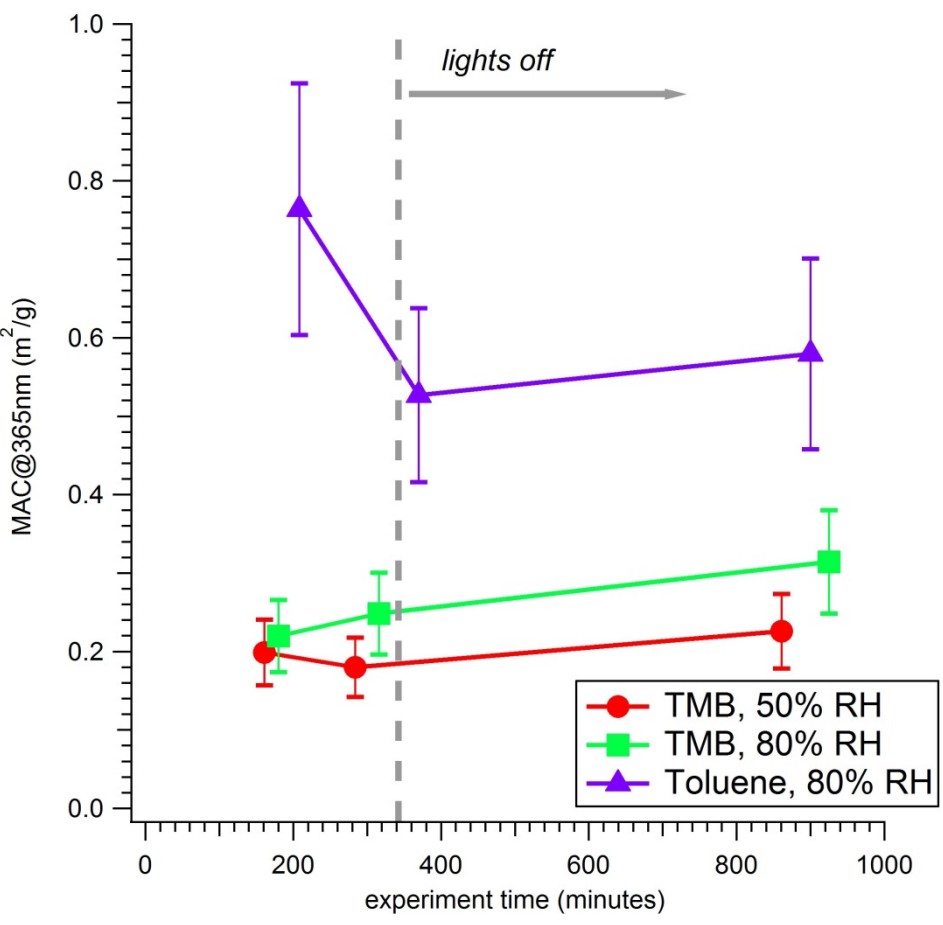


Figure 7. MAC values of aromatic SOA formed under high $NO_x$ conditions and aged in the chamber
with the lights off at different RH levels.
