# Peer review of "Optical Properties and Aging of Light Absorbing Secondary Organic 1 Aerosol 2 Jiumeng Liu1, Peng Lin2, Alexander Laskin2, Julia Laskin3, Shawn M. Kathmann3, Matthew Wise4, 3 Rvan Caylor4, Felisha Imholt4, Vane"

_Atmospheric Chemistry and Physics, 2016_

## Referee Comment (RC1) · Anonymous Referee #1 · 5 Jul 2016

This paper reports laboratory experiments aimed at understanding the formation of brown carbon (BrC) from various VOCs under different conditions. Studies on the evolution (aging) of the BrC formed are also presented. This work is a systematic analysis aimed at helping to interpret a number of ambient studies that showed BrC levels varied between cities with different mixtures of emissions.

The paper is highly relevant and interesting, it highlights the importance of anthropogenic SOA to BrC. I have only minor comments.

Are not many of the VOCs tested and attributed to anthropogenic SOA (eg, re discussions on urban SOA), also produced in biomass burning? If so, I would suggests the results have broader impacts than just what is discussed here.

Line 179: Suggest changing: which likely explains, to: which could explain,…. I really

don't know of ambient data supporting Lin et al (2014). For example, Washenfelder et al. (2015, Geophys. Res. Lett., 42, 10.1002/2014GL0624442015) saw no evidence that iepox (isoprene SOA) contributed to ambient BrC at a remote site in Alabama as part of SOAS where the aerosol is acidic (ie, papers show that it was acid catalyzed isoprene SOA, eg, see Xu et al, P. Natl. Acad. Sci., 112(1), 37-42, 2015).

Last line of section 3.1, (lines 228-230) I think ambient data that includes actual light absorption coefficients are need to make this statement. The logic of the line is unclear.

In section 3.4, the authors might also want to consider showing changes in the compete spectra, not just changes in absorption at 365 nm. This may prove useful when comparing to ambient data. This could go in supplementary material.

Line 372-373 regarding the discussion that alpha pinene and isoprene SOA produces little BrC. Again I would suggest the authors look at Washenfelder et al. As noted above, there is no evidence for isoprene SOA, but maybe pinene SOA from night-time reaction with NO3 radical.

Line 385, typo, ranged ?
* * *

---

## Referee Comment (RC2) · Anonymous Referee #2 · 8 Jul 2016

BrC has raised attention over the past decade because of its important contribution to light absorption and climate forcing. The authors aimed to unravel the complex links between SOA light absorption and chemical composition. To achieve this goal, biogenic VOCs (isoprene, alpha-pinene) and anthropogenic VOCs (TMB, toluene) were exposed to varying NOx levels in a chamber. The results show BrC formed from anthropogenic VOCs; specifically, the toluene SOA has the highest absorption under high-NOx level and 30-80% RH conditions. They found a nearly 50% underestimate of the MAC values based on a partitioning model, which is usually the case for current climate models. They also discussed the effect of RH and aging on the optical properties of aromatic BrC. In the end, the authors calculated the imaginary refractive indices and compared them with previous literature values. Overall, the experiments are described well and the results provide new insights into the BrC light absorption. I

favor its publication in ACP with the following minor revisions.

-in section 3.3, why is there no absorption enhancement when RH was increased from 30% to 80%?

-fig 1, the alpha-pinene SOA has a higher absorption than the isoprene SOA over the 300-350 nm wavelength range. It might be appropriate to comment on this. It will also be useful to list the MAC values in the supplement.

-fig 6, it's interesting that the authors observed a decline in the MAC values within 10 hours. I wonder whether the authors tried the 80% condition. And I suggest listing the change in MAC values over time in the supplement.

-please add a column in table 1 to label the high-NOx, low-NOx and NOx-free levels.

-line 178, delete (Lin et al., 2014b). Citation already given in line 176, be careful about repeating the citation, line 194, 220, 240, 242. . . . . .

-line 242 and line 412, there is some ambiguity. Is biomass burning anthropogenic or non-anthropogenic? In line 242 the authors see it as anthropogenic, in line 412, however, it is treated as non-anthropogenic.

-line 273 to line 275: I would suggest rephrasing the sentence.
* * *

---

## Author Comment (AC1) · 15 Sep 2016

We thank the reviewers for their constructive comments. Specific responses to each of the comments are provided below (reviews' comments in black, our responses in blue, and the manuscript text follows in italics with changes in bold).

**Anonymous Referee #1:**

This paper reports laboratory experiments aimed at understanding the formation of brown carbon (BrC) from various VOCs under different conditions. Studies on the evolution (aging) of the BrC formed are also presented. This work is a systematic analysis aimed at helping to interpret a number of ambient studies that showed BrC levels varied between cities with different mixtures of emissions.

The paper is highly relevant and interesting, it highlights the importance of anthropogenic SOA to BrC. I have only minor comments.

Are not many of the VOCs tested and attributed to anthropogenic SOA (eg, re discussions on urban SOA), also produced in biomass burning? If so, I would suggests the results have broader impacts than just what is discussed here.

The reviewer raises a good point. Other than primary BrC, which is directly emitted into the atmosphere from biomass burning, gas-phase emissions from burning are also exposed to sunlight and oxidants and can generate secondary products, including SOA. And as the reviewer suggested, nitrated aromatic compounds, identified in this work as strong light absorber in anthropogenic SOA, were also observed in some SOA samples produced from aging of biomass burning (e.g., Desyaterik et al., 2013; Iinuma et al., 2010). However, studies have also shown that SOA formation in biomass burning plumes is highly variable and dependent on factors such as fuel types, mass combustion efficiency, and aerosol aging, suggesting that the contribution of biomass burning SOA should be examined and parameterized carefully (Hennigan et al., 2011). To avoid "over selling" our results, we have carefully added some discussion regarding to the application of our results to biomass burning in lines 430-435, as shown below:

*Similar light-absorbing compounds have been identified in certain SOA samples originating from biomass burning (Desyaterik et al., 2013;Iinuma et al., 2010); since substantial variations in SOA formation in biomass burning plumes have been observed both chemically and physically due to fuel types and fire aging conditions (Hennigan et al., 2011), we cannot simply assume similar effects of those parameters on SOA produced from biomass burning emissions.*

References:
Desyaterik, Y., Sun, Y., Shen, X. H., Lee, T. Y., Wang, X. F., Wang, T., and Collett, J. L.: Speciation of "brown" carbon in cloud water impacted by agricultural biomass burning in eastern China, J Geophys Res-Atmos, 118, 7389-7399, Doi 10.1002/Jgrd.50561, 2013.
Hennigan, C. J., Miracolo, M. A., Engelhart, G. J., May, A. A., Presto, A. A., Lee, T., Sullivan, A. P., McMeeking, G. R., Coe, H., Wold, C. E., Hao, W. M., Gilman, J. B., Kuster, W. C., de Gouw, J., Schichtel, B. A., Collett Jr, J. L., Kreidenweis, S. M., and Robinson, A. L.: Chemical and physical

transformations of organic aerosol from the photo-oxidation of open biomass burning emissions in an environmental chamber, Atmos. Chem. Phys., 11, 7669-7686, 10.5194/acp-11-7669-2011, 2011.

Iinuma, Y., Böge, O., Gräfe, R., and Herrmann, H.: Methyl-Nitrocatechols: Atmospheric Tracer Compounds for Biomass Burning Secondary Organic Aerosols, Environ Sci Technol, 44, 8453-8459, 10.1021/es102938a, 2010.

Line 179: Suggest changing: which likely explains, to: which could explain,…. I really don't know of ambient data supporting Lin et al (2014). For example, Washenfelder et al. (2015, Geophys. Res. Lett., 42, 10.1002/2014GL0624442015) saw no evidence that iepox (isoprene SOA) contributed to ambient BrC at a remote site in Alabama as part of SOAS where the aerosol is acidic (ie, papers show that it was acid catalyzed isoprene SOA, eg, see Xu et al, P. Natl. Acad. Sci., 112(1), 37-42, 2015).

We have changed Line 179 as suggested by reviewer.

We felt that the Lin study deserved citation because it is an example of SOA derived from isoprene producing light-absorbing SOA, which is in contrast to our study. We note that our experiments were conducted under conditions that are expected to inhibit formation of SOA from the IEPOX pathway, in contrast to the Lin study (i.e., no acidic seed particles were used in our experiments). We can say with certainty that the different reaction conditions relative to Lin et al will produce different types of SOA (Liu et al., 2016). Because we inhibited formation from IEPOX, we are unable to make any conclusions about optical properties of IEPOX SOA from our data. Evaluation of the real-world impact of IEPOX SOA formation on aerosol optical properties is beyond the scope of our study.

References:
Liu, J., D'Ambro, E. L., Lee, B. H., Lopez-Hilfiker, F. D., Zaveri, R. A., Rivera-Rios, J. C., Keutsch, F. N., Iyer, S., Kurten, T., Zhang, Z., Gold, A., Surratt, J. D., Shilling, J. E., and Thornton, J. A.: Efficient Isoprene Secondary Organic Aerosol Formation from a Non-IEPOX Pathway, Environ Sci Technol, 10.1021/acs.est.6b01872, 2016.

Last line of section 3.1, (lines 228-230) I think ambient data that includes actual light absorption coefficients are need to make this statement. The logic of the line is unclear.

We have edited the sentence as below:

*Since fulvic acid is often cited as a surrogate of* **strong light-absorbing** *atmospheric BrC* **associated with biomass burning**, *this comparison shows that light absorption by BrC produced from anthropogenic VOCs can be significant under certain photochemical condition*, **consistent with high MAC values measured previously in urban environments when biomass burning impacts were low (e.g., Zhang et al., 2011, 2013; Liu et al., 2013)**.

In section 3.4, the authors might also want to consider showing changes in the compete spectra, not just changes in absorption at 365 nm. This may prove useful when comparing to ambient data. This could go in supplementary material.

Complete spectra are now provided in supplemental materials as Figure S4. We have added a description in lines 331-332 that: complete spectra in the wavelength range of 300-700 nm are provided in Figure S4.

Line 372-373 regarding the discussion that alpha pinene and isoprene SOA produces little BrC. Again I would suggest the authors look at Washenfelder et al. As noted above, there is no evidence for isoprene SOA, but maybe pinene SOA from night-time reaction with NO3 radical.

Discussion on pinene SOA from night-time reaction with NO3 radical has been added to lines 385-386. We have also edited lines 372-373 to emphasize that our statement refers only to the conditions we investigated in our experiments (see below). We unfortunately did not conduct any experiments investigating BrC formation from $NO_3$ oxidation of pinene.

Sentence at lines 372-373 (now lines 380-382) is edited as:

*Although α-pinene and isoprene have large contributions to the global SOA budget, they were shown to produce SOA with very small light absorption coefficients **under the photochemical conditions we investigated**, which agrees with literature data (i.e., Nakayama et al., 2010;Lang-Yona et al., 2010).*

Line 385, typo, ranged ?

The word "ranged" has been changed to "ranging".

---

## Author Comment (AC2) · 15 Sep 2016

We thank the reviewers for their constructive comments. Specific responses to each of the comments are provided below (reviews' comments in black, our responses in blue, and the manuscript text follows in italics with changes in bold).

**Anonymous Referee #1:**

BrC has raised attention over the past decade because of its important contribution to light absorption and climate forcing. The authors aimed to unravel the complex links between SOA light absorption and chemical composition. To achieve this goal, biogenic VOCs (isoprene, alpha-pinene) and anthropogenic VOCs (TMB, toluene) were exposed to varying NOx levels in a chamber. The results show BrC formed from anthropogenic VOCs; specifically, the toluene SOA has the highest absorption under high-NOx level and 30-80% RH conditions. They found a nearly 50% underestimate of the MAC values based on a partitioning model, which is usually the case for current climate models. They also discussed the effect of RH and aging on the optical properties of aromatic BrC. In the end, the authors calculated the imaginary refractive indices and compared them with previous literature values. Overall, the experiments are described well and the results provide new insights into the BrC light absorption. I favor its publication in ACP with the following minor revisions.

-in section 3.3, why is there no absorption enhancement when RH was increased from 30% to 80%?

That is a good question that we are unfortunately unable to definitively answer. In the second paragraph of section 3.3 (lines 297-317), we discuss possible reasons for this observation, though we can only present hypotheses that need further testing. We hypothesize that observations are related to particle-phase reactions because it is difficult to identify any gas-phase reactions involving water vapor that produce BrC chromophores. We assume that the modest to high RH serves as an on/off "trigger" in the production of light-absorbing compounds. We presume that water is not directly involved in these reactions so that additional condensed phase water provided by increasing RH from 30% to 80% does not further enhance reaction rates. Further investigations, including molecular analysis on SOA products formed under various RH conditions, would help answer this specific question.

-fig 1, the alpha-pinene SOA has a higher absorption than the isoprene SOA over the 300-350 nm wavelength range. It might be appropriate to comment on this. It will also be useful to list the MAC values in the supplement.

The text in lines 182-183 has been edited as:

*Compared to isoprene SOA,* *SOA formed from photochemical oxidation of α-pinene*  *showed slightly higher absorption in the 300-350 nm wavelength range, though the absolute MAC values are still small*.

The MAC values of isoprene, α-pinene, TMB and toluene SOA formed under $NO_x$-free and high-$NO_x$ conditions at 30% RH, i.e., data shown in Figure 1, are now listed in Table S1 in supplemental materials. A description has been added into Figure 1 caption:

*Figure 1. MAC values for SOA formed under NOx-free and high-NOx conditions, from isoprene, α-pinene, TMB, and toluene. Note the 10× difference in scale between the terpene and aromatic precursors.* **The MAC values shown in this figure are tabulated in the supplementary material (Table S1).**

-fig 6, it's interesting that the authors observed a decline in the MAC values within 10 hours. I wonder whether the authors tried the 80% condition. And I suggest listing the change in MAC values over time in the supplement.

We did conduct toluene SOA aging experiments at 80% RH though these experiments were conducted with a different experimental protocol. To be specific, UV lights were turned off after several hours of aging to distinguish between the effects of photolysis and hydrolysis. Results were shown in Figure 7. It is clear that the MAC values started to decrease in the first several hours, consistent with what we observed for the 30% RH condition shown in Figure 6.

In response to this comment and a similar comment from Reviewer 1, we added spectra showing the spectral changes as a function of time in Figure S4. We also added a table showing the MAC values in the 300-700 nm spectral range over time as Table S3.

-please add a column in table 1 to label the high-NOx, low-NOx and NOx-free levels.

We have added this column to Table 1.

-line 178, delete (Lin et al., 2014b). Citation already given in line 176, be careful about repeating the citation, line 194, 220, 240, 242: : :: : :

Repeating citations have been deleted.

-line 242 and line 412, there is some ambiguity. Is biomass burning anthropogenic or non-anthropogenic? In line 242 the authors see it as anthropogenic, in line 412, however, it is treated as non-anthropogenic.

The discussion at line 242 was mainly focused on the impacts of biogenic VOCs, specifically isoprene. To avoid confusion, lines 243-246 have been edited as:

*Hecobian et al. (2010) measured the light absorption of water-soluble organic carbon (WSOC) in Atlanta in different seasons and found that the winter WSOC has a ~3 times higher MAC than*

*summer, due to  **a higher fraction of organic aerosols formed from biogenic VOCs in summer** (Hecobian et al., 2010).*

-line 273 to line 275: I would suggest rephrasing the sentence.

The sentence is rephrased as:

*A third possibility is that organic peroxides and alcohols, which were shown to be the dominant component of isoprene SOA (Krechmer et al., 2015), may react with toluene SOA components to produce oligomers capable of absorbing in the UV/VIS that are not present in the single-precursor SOA particles.*